# A 26–28 GHz, Two-Stage, Low-Noise Amplifier for Fifth-Generation Radio Frequency and Millimeter-Wave Applications

**DOI:** 10.3390/s24072237

**Published:** 2024-03-31

**Authors:** Aymen Ben Hammadi, Mohamed Aziz Doukkali, Philippe Descamps, Constant Niamien

**Affiliations:** 1ASYGN (Analog System Design), 37 Rue Diderot Bâtiment B, 38000 Grenoble, France; 2ENSICAEN, Normandie Université, 14000 Caen, France; azizdoukkali14200@gmail.com (M.A.D.); philippe.descamps@ensicaen.fr (P.D.); 3ESIGELEC, Normandie Université, 76000 Rouen, France; constant.niamien@esigelec.fr

**Keywords:** fifth-generation mm-wave, low-noise amplifier, noise figure, cascode, SiGe BiCMOS

## Abstract

This paper presents a high-gain low-noise amplifier (LNA) operating at the 5G mm-wave band. The full design combines two conventional cascode stages: common base (CB) and common emitter (CS). The design technique reduces the miller effect and uses low-voltage supply and low-current-density transistors to simultaneously achieve high gain and low noise figures (NFs). The two-stage LNA topology is analyzed and designed using 0.25 µm SiGe BiCMOS process technology from NXP semiconductors. The measured circuit shows a small signal gain at 26 GHz of 26 dB with a gain error below 1 dB on the entire frequency band (26–28 GHz). The measured average NF is 3.84 dB, demonstrated over the full frequency band under 15 mA current consumption per stage, supplied with a voltage of 3.3 V.

## 1. Introduction

With the significant growth of fifth-generation (5G) millimeter-wave (mm-wave) applications, the performance of the radio front-end is increasingly becoming the critical element in most wireless receivers [1,2,3]. The emergence of process technologies, such as silicon–germanium (SiGe) BiCMOS, is potentially the most economical front-end solution, providing low cost and high performance [4]. In fact, NXP’s SiGe 0.25 µm process technology offers a high-performance mm-wave front-end solution with low noise characteristics, making it suitable for applications requiring high signal-to-noise ratios (SNRs), especially low-noise amplifiers (LNAs). In addition, SiGe transistors exhibit high cutoff frequencies (ft ≈ 180 GHz) and maximum oscillation frequencies (fmax≈ 200 GHz), allowing for the design of high-frequency circuits such as RF front-end modules, millimeter-wave systems, and microwave amplifiers.

Overall, SiGe 0.25 µm technology from NXP provides a compelling combination of high-speed performance, low noise, mixed-signal integration, high-frequency operation, low power consumption, and process compatibility, making it well-suited for a wide range of applications in communications, consumer electronics, automotive, aerospace, and industrial markets.

This paper presents the design and measurement of a low-noise amplifier, developed for the new generation of compact and bidirectional active antennas integrating a voltage amplifier instead of a conventional power amplifier (PA) for 5G applications. Such an amplifier is part of the transceiver, conditioning (filtering, amplifying) the signal in receiving or transmitting modes [5]. As an indispensable receiver block, the low-noise amplifier (LNA) significantly impacts the system’s performance. In fact, the convenient LNA is expected to provide a low noise figure (NF), good impedance matching (S11 < −10 dB), and relatively high-power gain based on the input matching network (MN), which is the essential part of the system [5,6]. 

In current knowledge, the front-end/back-end is designed separately from the antenna and packaged in the form of an integrated circuit, whose input/output (I/O) impedances are set close to 50 Ω (impedance antenna). Other chips’ impedances are not 50 Ω, requiring an external impedance-matching circuit. The advantage of this basic approach (Figure 1a) is that it gives rise to an almost ready-to-use front-end/back-end, whatever the characteristics of the antenna. However, this approach completely ignores the benefits of the electromagnetic behavior of the antenna.

For example, a half-wave dipole antenna in receive mode presents an almost constant open-circuit (no load) voltage over the entire frequency band below its natural resonance under a plane wave illumination. This attractive behavior can be exploited by adding a voltage amplifier, which would convert the maximum voltage captured into power from all frequencies below the resonance (Figure 1b).

We aim to design a new generation of active antennas integrating a bidirectional voltage amplifier (transmission/reception) for 5G applications, packaged as a chip mounted and directly interfaced with the antenna without impedance care.

The figure below depicts the full chip circuit. It contains four antennas operating at 26 GHz, four LNAs/Pas, a switch to select TX or RX transmission, and a DC bias circuit (Figure 2).

In the receiving-mode operation, this objective corresponds to the design of an LNA circuit in the 5G band, with a high input impedance (typically close to 150 Ω), which is higher than that of the receiving antenna, often near 50 Ω. This impedance mismatch establishes the compromise between the efficiency produced by the antenna and the noise figure.

The idea of designing an LNA is that it can be directly connected to the receiving antenna, for instance, a dipole, without conventional impedance matching. In this approach, as mentioned before, the LNA should have a higher input impedance than the receiving antenna (Z_in_LNA_ >> Z_in_Antenna_) to set the antenna in the open-circuit operating mode, where it receives incident waves maximally. Meanwhile, the LNA’s output should match the output load (Z_out_LNA_ = Z_Load*_) to maximize the power transfer. Other benefits of the voltage-controlled LNA are antenna size reduction and broadband operation under a controllable gain.

Since a single-stage design is inefficient in achieving the required design trade-offs in the mm-wave band, most LNA circuits include multiple stages [7,8], like the cascode topology [9,10,11,12,13]. With the common-base transistor, this design has better stability and reverse isolation, as well as improved bandwidth and high gain over the entire mm-wave frequency band. In addition, the cascode topology’s output admittance is quite low compared with the common-emitter topology, based on its low capacitive component [8].

This paper presents a classical two-stage design of a 26–28 GHz LNA based on 0.25 μm SiGe BiCMOS technology. As usual, the emitter degeneration inductor is used to obtain noise and input impedance matching simultaneously. The paper is organized as follows. Section 2 describes the design methodology and the different detailed parts of the LNA design. Simulation and experimental results are presented in Section 3, followed by a comparison with the state of the art on LNAs. Finally, conclusions are drawn in Section 4.

## 2. Modeling and Circuit Design

Before starting the design of the LNA circuit, we describe the design techniques used for LNA circuit design. The design of each block of LNA is described in detail in the following section.

### 2.1. Device Size and Bias Selection

Transistor size is the first important step in designing an active integrated circuit [14]. The main design specifications of the proposed LNA include a gain of 30 dB and an NF below 2.5 dB in the frequency band between 26 GHz and 28 GHz, which fits 5G antenna transceivers. Simulating the DC and RF characteristics of the transistor devices is the first step to verify the feasibility of a low-noise amplifier. For example, the DC characteristics, NF, NFmin, and gain as a function of the base-emitter voltage of the transistor (Vbe) are presented in Figure 3. We chose a 0.25 μm BiCMOS SiGe process with an f_T_/f_MAX_ ratio of around 180/200 GHz.

Thus, we simulated the bipolar transistor( dotted in red) performance in the mm-wave frequency range (26–28 GHz) and deduced the optimized emitter length, emitter width, and number of emitters. The transistor size was chosen to minimize the noise figure (NF) while maintaining a power consumption that was as low as possible.

The HBT transistor was biased at 3.3 V with a current consumption of 8 mA to achieve the minimum NF. The emitter width and length were set to 0.4 um and 1.5 um to match the specifications of current consumption, NF, and gain. The number of emitters was set to N = 4 to have the optimum base resistance $r_{b}$. 

Based on Equation (1), the transistor size was scaled to have the optimal transconductance $g_{m}$, minimum $r_{b}$, and higher current gain β and transition frequency $f_{t}$.
(1)$${NF}_{min}=1+\frac{1}{\beta }\sqrt{2g_{m}r_{b}}\sqrt{\frac{1}{\beta }+{\left(\frac{f}{f_{T}}\right)}^{2}}$$

With this given configuration, as shown in Figure 4, the transistor could provide a small signal gain of 15.44 dB and a minimum NF of 1.82 dB under a bias voltage ($v_{be}$) of 0.86 V.

Note that the size and polarization of the BJT transistor will be optimized in the final phase of design according to the desired performances.

In summary, transistor size (which involves selecting the appropriate dimensions (W and L)) as well as biasing (which affects its intrinsic gain and bandwidth) play crucial roles in determining the gain characteristics of an amplifier circuit, especially in high-frequency applications where gain roll-off can occur. It is for this reason that a great intention was taken on this part to lead the circuit to good performance. In fact, a large transistor can typically offer higher gain but may have limited bandwidth due to parasitic capacitances. So, by carefully selecting transistor sizes and biasing conditions, we could optimize the gain-bandwidth product of the LNA to minimize gain roll-off and achieve the desired performance across the operating frequency range.

### 2.2. Input-Matching Strategy and Inductive Emitter Generation

The input matching of an LNA is a critical part of noise performance. To obtain the lowest possible loss and consequently the lowest possible noise performance, an input-matching network consisting of LC components was used to transform the 150 Ω source impedance to an optimum noise source impedance Z_op,noise_ in the first gain stage. A degenerated common emitter (CE) technique was used to facilitate impedance matching.

Figure 5 shows an example of the equivalent circuit of a wideband input-matching network. Note that the following circuit was used just to analyze the input impedance of LNA as a function of the parameters of the circuit. The parasitic pad capacitance C_pad_ should be considered as a part of the matching network at mm-wave frequencies. 

Where $Z_{in}$ is the equivalent input impedance of the LNA and $Z_{b}$ is the input impedance of the bipolar transistor with degeneration inductor.

According to the small-signal analysis, the input impedance of the LNA can be written as follows:(2)$$Z_{in}(s)=\frac{s^{3}\left(R_{q}L_{b}C_{p}C_{be}\right)+s^{2}\left(L_{e}+L_{b}\right)C_{be}+sR_{q}C_{be}+1}{sC_{T}\left(s^{2}\frac{C_{be}}{C_{T}}\left(L_{e}C_{p}+L_{b}C_{pad}\right)+s\frac{C_{be}}{C_{T}}R_{q}C_{p}+1\right)}$$
where “s” is pulsation (rd/m) and
(3)$$C_{T}=C_{P}+C_{pad}+C_{be}$$

The equivalent resistance $R_{q}$ provides the required real part for the 150 Ω input matching (note that $L_{e}$ is the degeneration inductor, $\omega_{T}=2*\pi *f_{t}$ is the transition angular frequency, and $f_{t}$ is the transition frequency of the transistor). $L_{b}$ is used to enhance the quality factor and to keep the overall NF low. According to the Friis equation [14], the first stage mainly determines the NF. Meanwhile, the first CE stage can suppress the noise voltage of the inter-stage matching network. Therefore, the noise performance is like El-Nozahi’s solution, which has been well analyzed in [15].

Note that the following design considers the connection aspect of the chip to the antenna with wire bonding, whose characteristics have been included in the design modeled by wire-bonding inductance $L_{b}$.

As a conclusion for this part, impedance matching is essential for maximizing power transfer between stages of an amplifier circuit and ensuring efficient signal propagation.

In this LNA circuit, components in the matching network (inductors and capacitors) are selected and tuned to provide the desired impedance transformation over the operating frequency range of the amplifier. To avoid all problems coming from the matching network, techniques like Smith chart analysis, impedance-matching networks (L section, T section or pi-section), and iterative optimization algorithms are employed to design matching networks that achieve good impedance matching across the band and minimize reflection losses.

### 2.3. Topology Selection

In the mm-wave band, as a single-stage design is inefficient in meeting the expected design trade-offs, most LNA circuits are built using the multi-stage approach. Furthermore, the cascode topology fits well in realizing expected design goals by combining the advantages of common emitter and common base configurations [16]. The basic idea behind a cascode amplifier is to overcome the limitation of the miller effects due to intrinsic collector-base capacitance in the CE stage by adding a CB stage at its output, as shown in Figure 6.

The input resistance and the large transconductance of a CE amplifier are combined with the current-buffering property and superior high-frequency response of the common-base circuit to reduce the Miller effect [17]. Moreover, due to the common-base transistor, this topology exhibits better stability, high reverse isolation, improved bandwidth, and high gain over the entire mm-wave frequency band. As highlighted by [18], the output admittance of the cascode topology is quite low compared to the common-emitter topology because it has a low capacitive component.

The small-signal equivalent diagram corresponds to the structure shown in Figure 7.

The CE stage in cascode amplifier is critical since the NF of this stage determines the overall NF of the amplifier. Furthermore, input impedance determines the input matching network. The input impedance seen looking into the base of the CE stage is given by Equation (4).
(4)$$Z_{in}=R_{b}+R_{e}+\omega_{T}L_{e}+j\omega L_{e}+\frac{1}{\left(\frac{j\omega C_{\pi }}{1+g_{m}R_{e}}\right)}$$

For the system requirements, as the source impedance for the circuit design is 150 Ω, the imaginary part gets canceled out, and the real part is given by the following:(5)$$L_{e}=\frac{150-\left(R_{b}-R_{e}\right)}{\omega_{T}}\left(\mathrm{with}\;\omega_{T}=\mathrm{transition}\mathrm{pulsation}\right)$$where “$\omega_{T}$ = $2*\pi *f_{t}$” is the transition pulsation, and f_T_ is the transition frequency of the transistor. 

The NF small signal analysis results in the expression given in Equation (6). From this equation, it is clear that NF mainly depends on dominant contributions from the base resistance and the collector current, along with the internal resistance of the input source. Theoretically, the base resistance can be indefinitely reduced by increasing the emitter area using multi-emitter transistors or placing many transistors in parallel. In practice, this also increases the base-collector capacitance, limiting the lowest NF value that can be achieved.
(6)$$F=1+\frac{R_{b}+R_{e}}{R_{s}}+\frac{1}{2\beta r_{e}}\frac{{\left[R_{s}+R_{b}+R_{e}\right]}^{2}}{R_{s}}+\frac{r_{e}}{2R_{s}}{\left[\frac{{\beta r}_{e}+R_{s}+R_{b}+R_{e}}{\beta r_{e}}\right]}^{2}$$

### 2.4. Circuit Design

The gain of a single-stage amplifier is too small to obtain a 30 dB linear gain. Therefore, we chose two cascode stages, as illustrated in Figure 8. 

The conceptual schematic diagram of the realized Ka-band LNA is composed of two cascode amplifier stages. The first is optimized for noise and input matching, while the second is for gain/linearity, keeping minimal current consumption. Both active stages (transistors Q_1_, Q_2_, Q3, and Q_4_) have identical sizes, each using a single supply voltage of 3.3 V.

The first stage, composed of transistors Q_1_ and Q_2_, performs both input impedance matching at 150-Ω and noise matching. In contrast, the second stage (Q_3_, Q_4_) improves the amplifier’s voltage gain and provides an output impedance matching of 50 Ω. An emitter length of L = 1.5 µm, width of W = 0.4 µm, and fingers of N = 4 was chosen for transistors Q_1_ and Q_2_ to exploit the peak f_max_ of the technology. Standard base transistors were selected as two times the same size of CEs to increase the gain of the first stage by increasing the output impedance. Each cell was biased at 3.3 V. 

The input matching network composed of inductances L_1_ and L_2_ and capacitance C_1_ was used to transform the 150 Ω source impedance to the optimum noise source impedance Z_op,noise_ in the first gain stage. The inductance L_2_ was parallel to the ground and also had the function of electrostatic protection. Inductive source degeneration was applied in the first stage of the LNA to set the optimum noise impedance Z_op,noise_ and power impedance Z_op,power_ close to each other under simultaneous matching conditions. The output matching was obtained through the inductances L_5_ and L_6_, and capacitances C_5_–C_6_ to reach the flat gain and the broadband matching. The equivalent inductance of the bonding wires, used to connect the chip to the antenna, was considered in the design.

It is crucial to highlight that all matching networks are crafted with the purpose of broadening the bandwidth of the LNA. This serves to prevent the occurrence of gain roll-off, which leads to a decline in gain at elevated frequencies.

Also, obtaining consistent gain across the entire frequency spectrum poses a significant challenge at 26 GHz, mainly due to the influence of parasitic elements, substrate effects, and process variations. To address this challenge, we explored a few techniques for enhancing gain flatness like the optimization of layout configurations to mitigate the impact of parasitic capacitances and inductances.

Moreover, the influence of process variations becomes more pronounced, resulting in fluctuations in device parameters and performance metrics like gain, noise figure, and linearity. To address this challenge, we explored some strategies to alleviate the impact of process variations. This may involve the utilization of statistical design methodologies, the optimization of layout techniques to reduce substrate coupling and parasitic effects, and implementation of adaptive biasing schemes to counteract variations in device characteristics.

### 2.5. Simulation Results

A two-stage cascode LNA was designed using SiGe BiCMOS technology. Bear in mind that the main objective of this paper was to demonstrate the feasibility of a cascode LNA operating in the ka band with an input impedance equal to 150 Ω. Therefore, before starting the simulation, let us set the involved parameters.

With an adequately degenerated inductance, Figure 9 shows the position relationship between the circle of NF and the input impedance Z_in_ at 27 GHz. Even though 50 Ω is the reference value, 150 Ω is the real value. 

When the input source impedance Rs was 150 Ω, the NF was about 2.3 dB. This value was used in this work to obtain the best compromise between high input impedance and a low noise figure.

Once the input resistance was set, the entire LNA structure, including interconnections (such as wire bonding), was simulated using Spectre RF circuit simulator. Note that all simulation results presented below are extracted from post-layout.

The forward gain is an important parameter that influences the LNA’s power gain and voltage gain. In our targeted band of frequency between 26 GHz and 28 GHz, the obtained S_21_ lay between 21 dB and 23 dB on the useful frequency band. The forward gain of 22.8 dB was available at 27 GHz.

Bear in mind that the full LNA consumes 15 mA per stage. Both stages had the same transistor size, except that they were optimized in two different ways. The first one was optimized for the noise figure, while the second one was optimized for gain. Also note that increasing bias current could increase gain and noise figure, but it decreased the stability condition.

Figure 10 shows the input return loss was lower than −12 dB at frequencies between 26 to 28 GHz. The output return loss was lower than −13.5 dB all over the targeted band of frequency. At the center frequency, the input and output return losses dropped to −13.46 and −17.93 dB, respectively. 

As presented in Figure 11, NF and NF_min_ are plotted as a function of frequency. The obtained NF at 27 GHz was 3.84 dB, while NF_min_ was 3.74 dB. 

The proposed LNA was unconditionally stable, and the lowest value of Rollet’s stability factor (Kf) was around 5.54, as shown in Figure 12.

The input-referred 1 dB compression point of the proposed LNA was −16 dBm, as illustrated in Figure 13.

The third-order intercept point IIP3 of the proposed LNA was −12.17 dBm, as illustrated in Figure 14. The lines in blue and red are the fundamental output power and the third order intermodulation distortion products power (IMD3 or IM3), respectively.

While linearity remains a crucial factor in low-noise amplifier (LNA) design, the primary challenge in this particular design lies in achieving optimal gain and impedance matching. This entails the creation of an LNA capable of operating with a high input impedance of 150 Ohms.

Before moving on to the next part, we want to remind readers that process variations in semiconductor manufacturing can lead to variations in device parameters such as threshold voltage, channel length, and oxide thickness, affecting the performance of amplifier circuits. So, for this, common layout techniques include using dummy devices to balance process-induced variations, symmetric layout structures to minimize gradients in substrate biasing, and guard rings to reduce leakage currents and substrate coupling effects. All of these techniques lead to the mitigation of the impact of process variations and ensure consistent performance across manufacturing lots.

## 3. Measurements and Discussion

The fully integrated mm-wave LNA for 5G applications was fabricated in 0.25 μm SiGe BiCMOS technology. The chip layout is shown in Figure 15, and it occupies an area of 702 × 1220 µm^2^.

On-wafer measurements were performed on a standard substrate thickness chip of 150 μm. The circuit operated with a bias voltage of 3.3 V, while the measured DC current consumption was 15 mA per stage. 

Note that the circuit was composed of two identical stages.

RF wafer probes were used to measure the input and output ports. The measurement of S-parameters was carried out with a single-ended two-port VNA. Figure 16 shows the test chip of the measured LNA.

The simulated and measured S-parameters of the LNA presented in Figure 17a,b are in good agreement. As can be seen, the LNA achieved a maximum gain of 26 dB at 27 GHz and a gain greater than 25 dB over the 26–28 GHz frequency range. In addition, isolation S_12_ was less than −32 dB over the same frequency range. 

The minimum input return loss was S_11_ = −0.94 dB at 26 GHz and remained below 0 dB over the entire frequency band. This indicates that the LNA was stable. The output matching S_22_ exhibited a matching level less than −20 dB in the entire frequency band (S_22_ = −14.4 dB@27 GHz). The S_11_ degradation was likely due to the input impedance mismatch near 150 Ohm instead of 50 Ohm.

Note that dotted lines are the measurement results while solid lines are PLS simulation results.

The measured NF was around 2.35 dB at 27 GHz and 2.5 dB at 26 GHz, respectively, as shown in Figure 18. Note that dotted lines are the PLS simulation results while solid lines are measurement results.

The minimum NF measured was 2.33 dB@26 GHz and 1.77 dB@27 GHz. These values corresponded to the measurement at the chip level. The NF measured was much lower than its value in the event of assembly with connecting wires, which was 3.84 dB in the simulation. The effect of the package explains this difference. Indeed, the simulations considered 12 bonding wires corresponding to an inductance of 250 pH each, while the measurements used calibration and de-embedding directly on the wafer. The inductance of the bonding wires, especially to the ground, significantly degraded the noise figure by changing the optimum noise impedance. Also, the amplifier’s source impedance was 50 Ohms under on-wafer measurements, while the LNA was designed for a source impedance of 150 Ohms to operate as a voltage amplifier.

A comparison focusing on recent BiCMOS LNAs is tabulated in Table 1.

The developed LNA showed close gain and similar NF values to other circuits. Moreover, this work stands out by having the highest gain per stage, which is essential for low power consumption. The linearity of the LNA was simulated from the extraction of the compression point P1 dB and the point of intermodulation of order 3, IIP3. The simulations of the P1 dB and IIP3 at 27 GHz are provided in Figure 19 and Figure 20, respectively.

Other measurement results for LNA such as P1 dB and IP3 are not available. 

## 4. Conclusions

In this paper, a 26–28 GHz, two-stage LNA design for mm-wave 5G applications has been proposed. The novelty of this design is its operation as a voltage amplifier with a higher input impedance near 150 Ω instead of 50 Ω, as usual. This LNA behavior allows for the voltage picked up by the antenna to increase. The LNA was fabricated using 0.25 µm SiGe BiCMOS technology. Measurements showed that the LNA achieved a peak gain of 26 dB between 26 and 28 GHz with an NF of 2.45 at 26 GHz. This chip consumed 15 mA per stage under a 3.3 V supply.

## Figures and Tables

**Figure 1 sensors-24-02237-f001:**
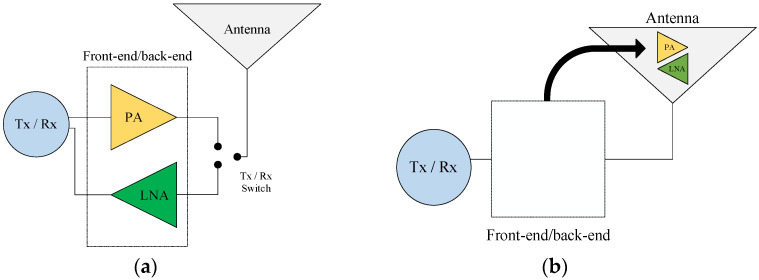
(**a**) Schematic of a basic approach simplifying the architecture of a transmission system. (**b**) Schematic of the proposed approach for the architecture of a transmission system.

**Figure 2 sensors-24-02237-f002:**
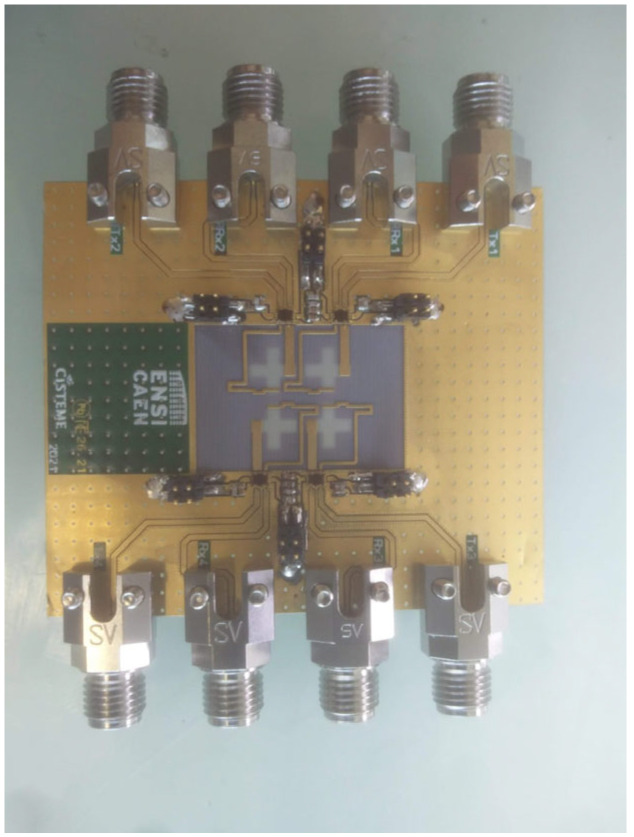
Full chip circuit for antenna with LNA/PA.

**Figure 3 sensors-24-02237-f003:**
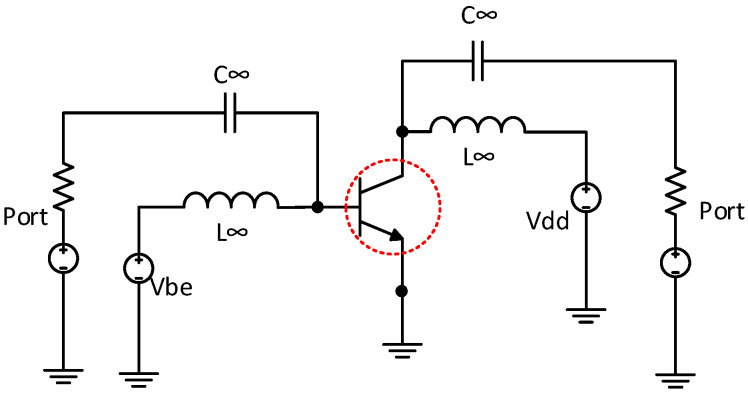
Circuit setup for optimum transistor size selection.

**Figure 4 sensors-24-02237-f004:**
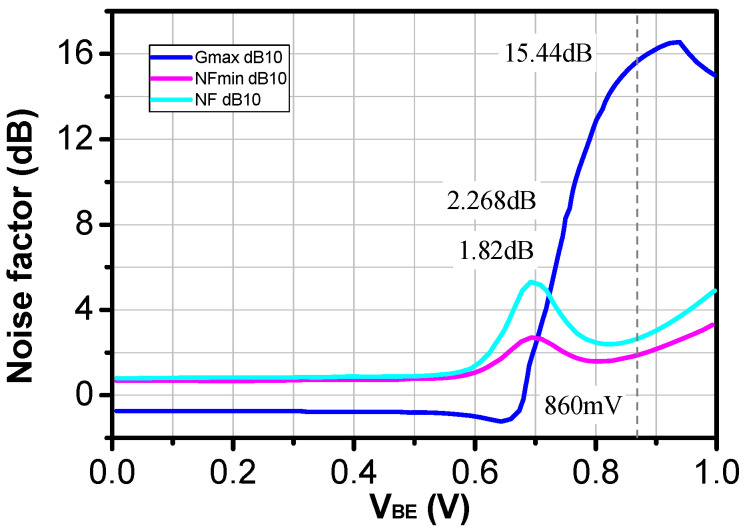
Characteristics of a transistor (gain. NF. and ${NF}_{min}$ versus $v_{be}$).

**Figure 5 sensors-24-02237-f005:**
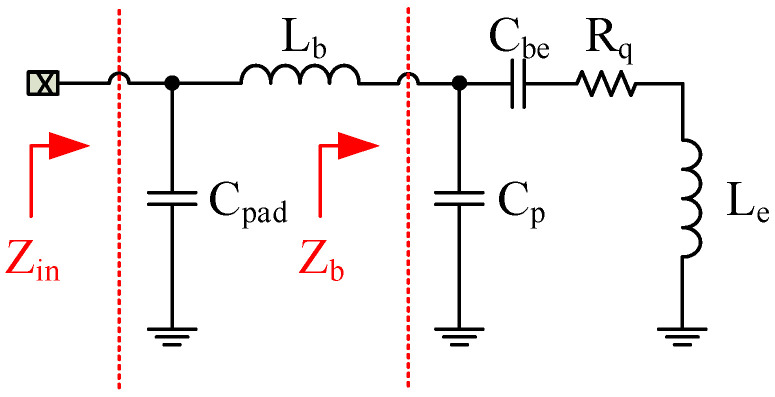
Input matching network.

**Figure 6 sensors-24-02237-f006:**
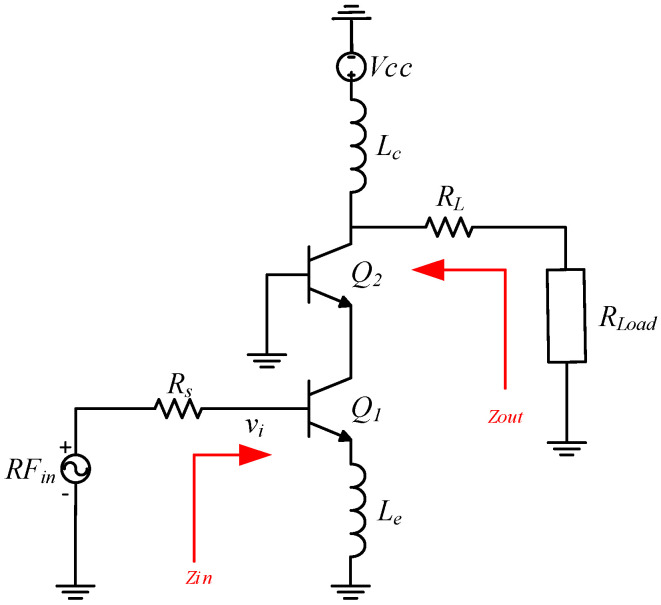
Cascode amplifier with emitter degeneration without biasing.

**Figure 7 sensors-24-02237-f007:**
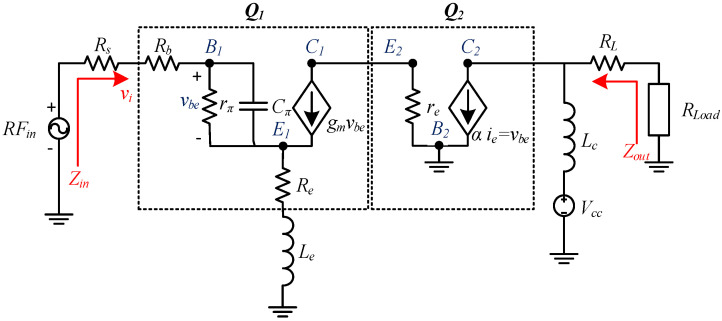
Simplified small signal hybrid π-model of cascode amplifier with emitter degeneration.

**Figure 8 sensors-24-02237-f008:**
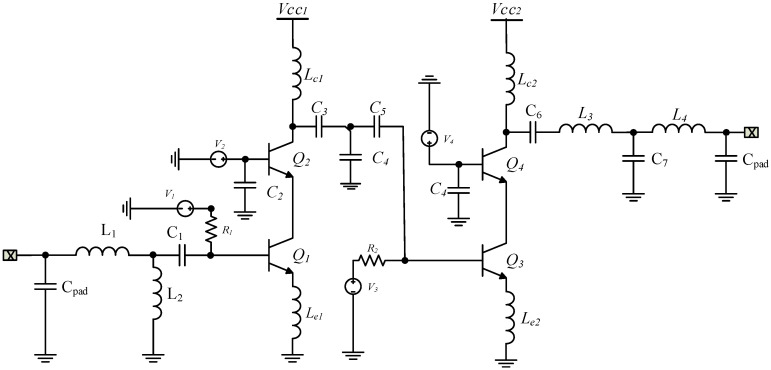
Schematic of proposed ka-band 5G LNA.

**Figure 9 sensors-24-02237-f009:**
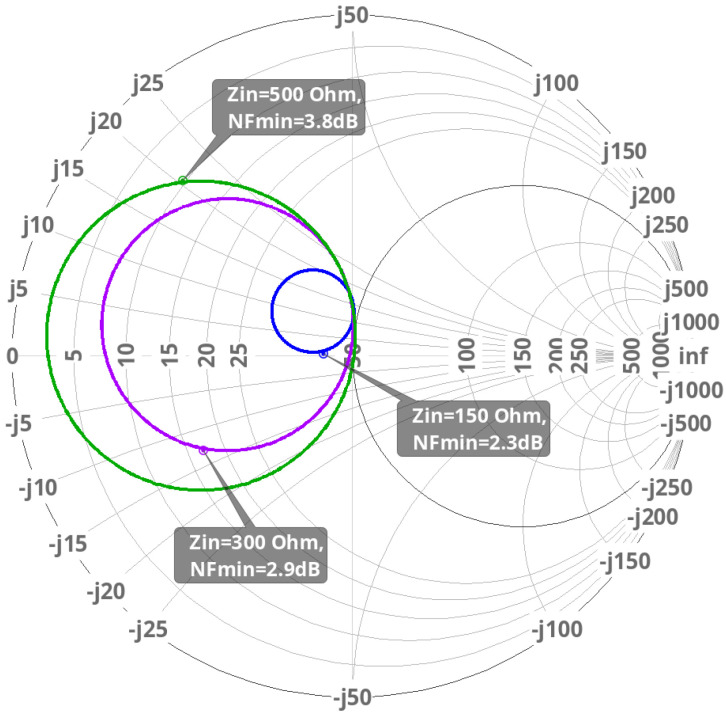
Circle of noise figure and the input impedance at 27 GHz.

**Figure 10 sensors-24-02237-f010:**
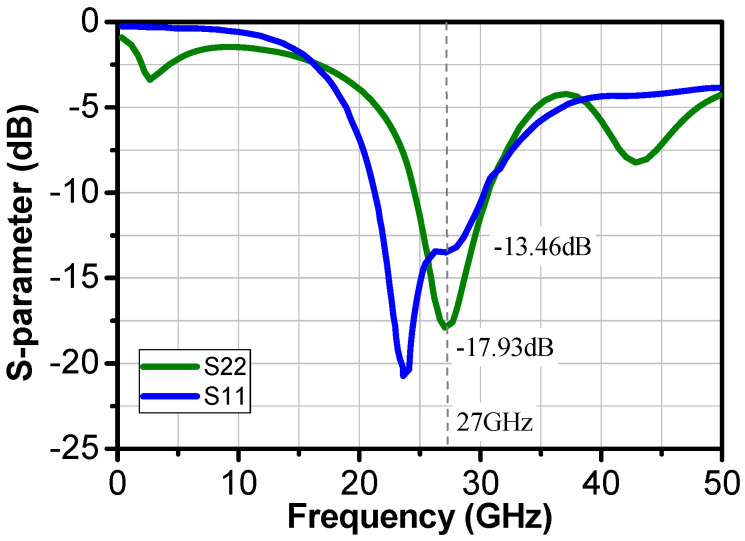
Simulation results of the input and output return losses of the two-stage LNA.

**Figure 11 sensors-24-02237-f011:**
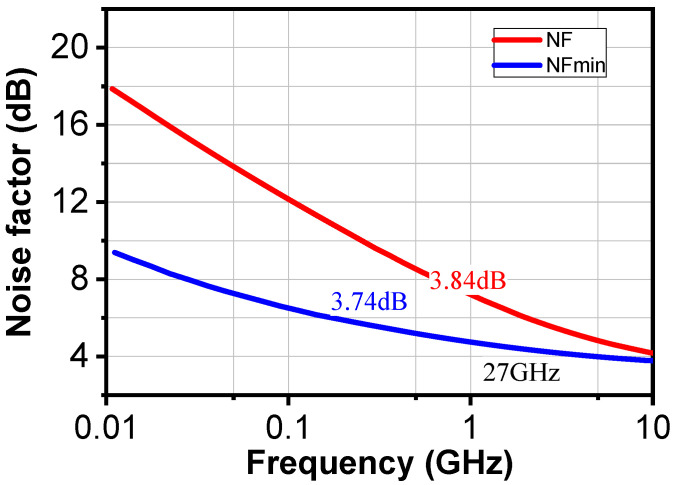
Simulated noise figure of the two-stage LNA.

**Figure 12 sensors-24-02237-f012:**
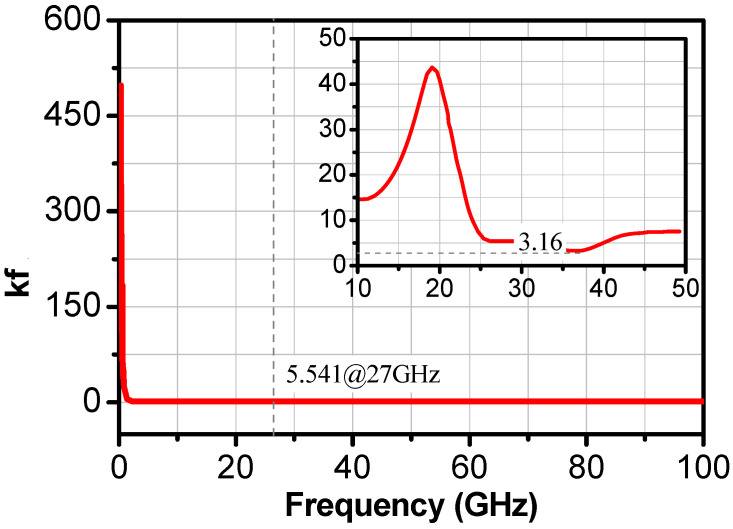
Stability factor (Kf) of the two-stage LNA.

**Figure 13 sensors-24-02237-f013:**
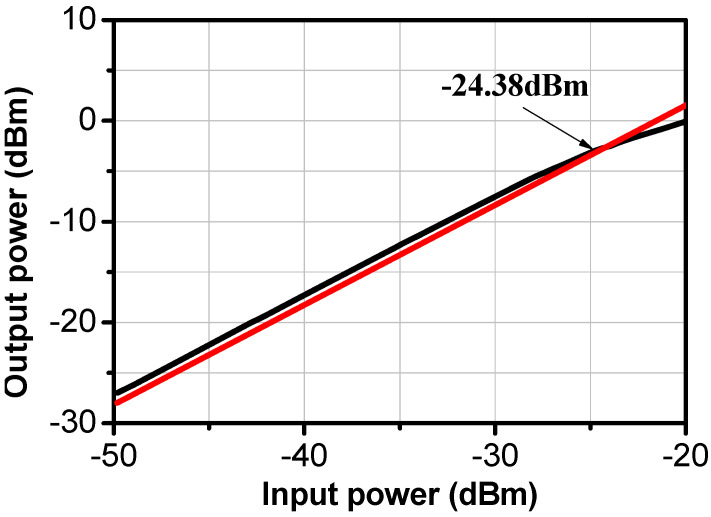
Input-referred P1 dB of the proposed LNA.

**Figure 14 sensors-24-02237-f014:**
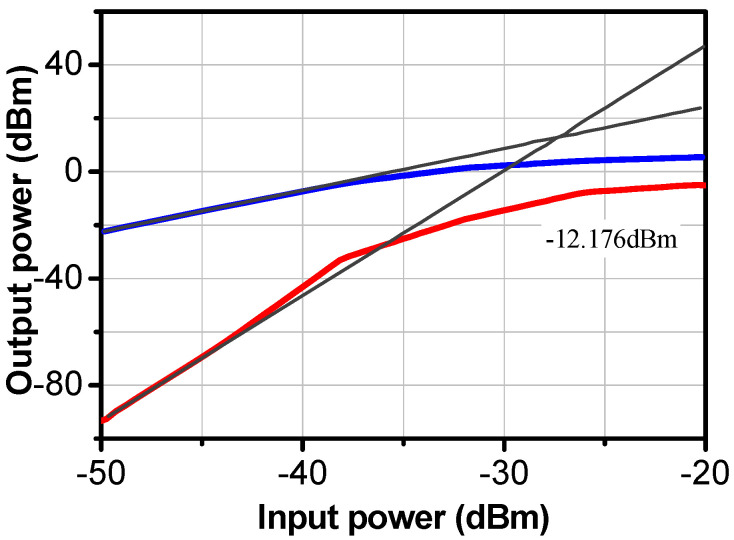
IIP3 of the proposed LNA.

**Figure 15 sensors-24-02237-f015:**
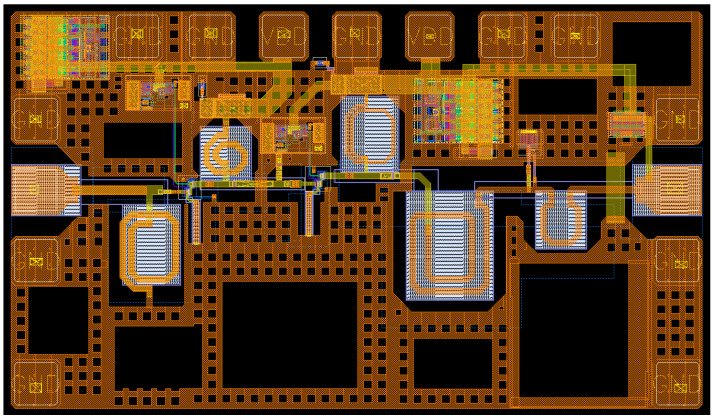
Layout and die photo of the fabricated 5G mm-wave LNA.

**Figure 16 sensors-24-02237-f016:**
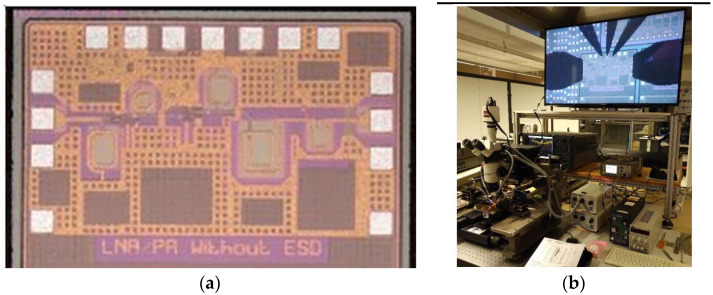
LNA under test: (**a**) Photograph of the LNA chip. Dimensions are 702 × 1220 µm^2^. (**b**) Measurement bench for the characterization of the LNA.

**Figure 17 sensors-24-02237-f017:**
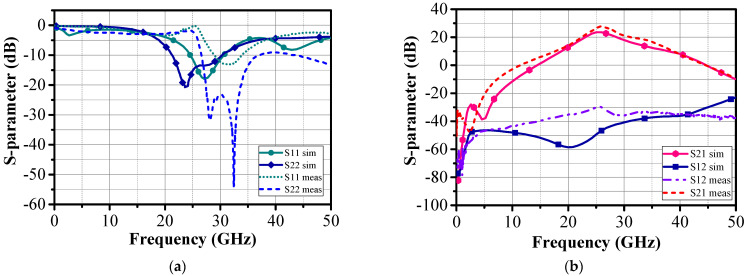
Simulated and measured values of S_11_, S_22_, S_21_, and S_12_ for the two-stage LNA.

**Figure 18 sensors-24-02237-f018:**
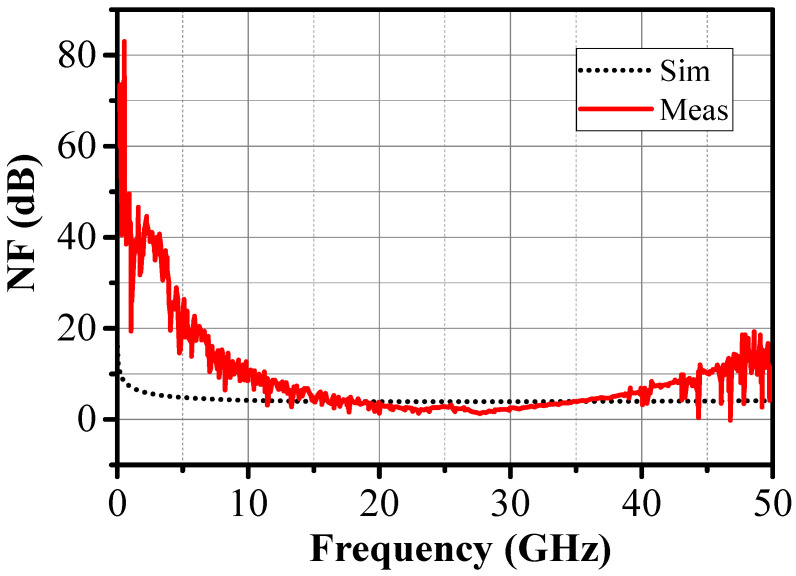
Simulated and measured NF for the two-stage LNA.

**Figure 19 sensors-24-02237-f019:**
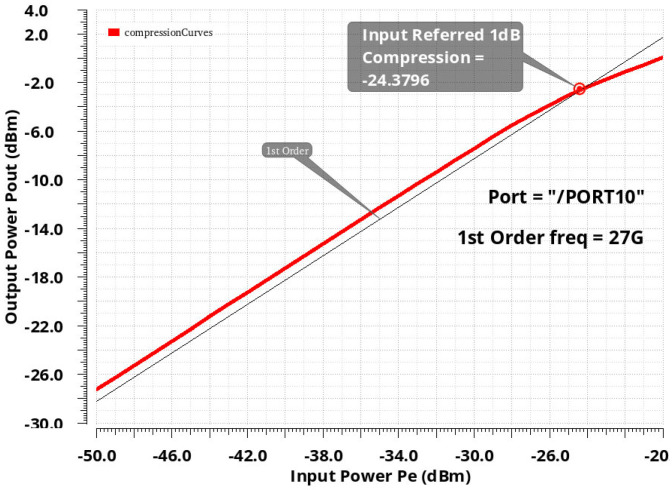
Input_referred P1 dB of the proposed LNA.

**Figure 20 sensors-24-02237-f020:**
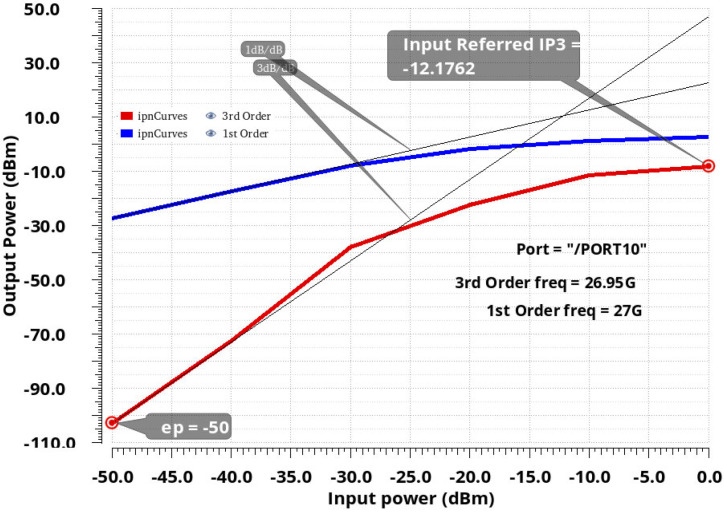
IIP3 of the proposed LNA.

**Table 1 sensors-24-02237-t001:** Performances of BiCMOS LNAs.

	[18]	[19]	[20]	[1]	[21]	[4]	This Work
Process	0.13 µm BiCMOS	0.25 µm BiCMOS	0.13 µm CMOS	0.18 µm BiCMOS	0.13 µm BiCMOS	0.25 µm BiCMOS	0.25 µm BiCMOS
V_cc_ (V)	1.5	1.8	1.2	1.8	1.2	2	3.3
S21 (dB)	14.5	10.5	22.14	18.6	22.2	28.5	26
f_0_ (GHz)	24	16–43	27–31	22–32.5	22–47	29–37	26–28
NF (dB)	2.7	2.5–4	1.86	5.5	3–4.3	3.1–4.1	2.35
P_dc_ (mW)	10	24	33.4	5	9.5	80	30
P_out,dB_ (dBm)	−12	−8	−10	−14.6	−23	−23	NA *
IIP3 (dBm)	N/A	1.8	−16	−5.7	−13.8	−12.5	NA *

NA *: not available for measurement. Simulation results are available in the text.

## Data Availability

The data used to support the findings of this study are confidential.

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
