# Peer review of "A 26–28 GHz, Two-Stage, Low-Noise Amplifier for Fifth-Generation Radio Frequency and Millimeter-Wave Applications"

_sensors, 2024, doi:10.3390/s24072237_

Round 1

Reviewer 1 Report

Comments and Suggestions for Authors

The authors presented “A 26-28GHz, Two-stage Low Noise Amplifier for 5G RF & mm-Wave Applications”. The paper is well-designed and presented LNA for 5G mm-wave applications. However, the authors can further strengthen their work and make it even more valuable to the research community. Here are some comments to address:

·         State of the art in the introduction is very limited.

·         It needs to discuss the linearity of the amplifier, which is an important parameter for 5G applications. A clear explanation is needed on how the chosen design choices (e.g., topology, transistor sizing, biasing) to influence linearity. Also, it could be compared with other LNAs in the literature.

·         The authors need to discuss more on specific challenges like increased gain roll-off, reduced gain flatness, and increased sensitivity to process variations at mm-wave frequencies.

·         Please explain how transistor sizing and biasing impact gain roll-off, how the matching network components ensure good impedance matching across the band, and how layout techniques mitigate process variations.

·         Provide more details on the integration of the LNA with active antennas

·         While presenting total current draw, a breakdown of power consumption per stage and its impact on gain and noise performance would be insightful.

·         Most of the time authors have used the word called “project”. Please remove it and rephrase those sentences.

Author Response

Please find attached a reply for all comments

Reviewer 2 Report

Comments and Suggestions for Authors

In this article, the authors make the case for an LNA operating with a higher input impedance with respect to the classic 50 ohms for better integration with the antenna. Overall, this appears to be a comprehensive design, encompassing fabrication and experimental results. The design is compared with literature; however, it should be noted that the actual novelty in the design procedure seems to be somewhat limited.

The authors could enhance the work by providing more details about the technology employed. Why choose 0.25μm-SiGe BiCMOS technology over others?

Additionally, the Introduction discusses a project called "ACTIS," but no reference/link is provided.

Finally, could the authors better explain what they mean by: "active antennas integrating a bidirectional voltage amplifier (transmission/reception) packaged in the form of a chip mounted directly in the antenna’s footprint"? Do they mean a chip integrating PA, LNA, and switch? From a graphical perspective, I suggest merging Figs 1 and 2 into a single Fig, with (a) and (b) cases.

Comments on the Quality of English Language

I suggest a thorough English review before publication. 

Author Response

(The authors gave the same response as above.)

Round 2

Reviewer 1 Report

Comments and Suggestions for Authors

The authors addressed all my comments. It may be considered for publication with the consideration of other reviewer(s) recommendations if any.